# Dosimetry Effects Due to the Presence of Fe Nanoparticles for Potential Combination of Hyperthermic Cancer Treatment with MRI-Based Image-Guided Radiotherapy

**DOI:** 10.3390/ijms24010514

**Published:** 2022-12-28

**Authors:** Amiel Gayol, Francisco Malano, Clara Ribo Montenovo, Pedro Pérez, Mauro Valente

**Affiliations:** 1Instituto de Física E. Gaviola (IFEG), CONICET & Facultad de Matemática, Astronomía, Física y Computación (FAMAF), Universidad Nacional de Córdoba, Ciudad Universitaria, Córdoba 5000, Argentina; 2Laboratorio de Investigación e Instrumentación en Física Aplicada a la Medicina e Imágenes por Rayos X (LIIFAMIRx), Facultad de Matemática, Astronomía, Física y Computación (FAMAF), Universidad Nacional de Córdoba, Ciudad Universitaria, Córdoba 5000, Argentina; 3Centro de Excelencia de Física e Ingeniería en Salud (CFIS), Departamento de Ciencias Físicas, Universidad de La Frontera, Av. Salazar 01145, Casilla 54D, Temuco 4811230, Chile

**Keywords:** convergent beam radiotherapy, CONVERAY system, Monte Carlo simulation, patient-specific dosimetry, stereotactic radiosurgery, lung cancer

## Abstract

Nanoparticles have proven to be biocompatible and suitable for many biomedical applications. Currently, hyperthermia cancer treatments based on Fe nanoparticle infusion excited by alternating magnetic fields are commonly used. In addition to this, MRI-based image-guided radiotherapy represents, nowadays, one of the most promising accurate radiotherapy modalities. Hence, assessing the feasibility of combining both techniques requires preliminary characterization of the corresponding dosimetry effects. The present work reports on a theoretical and numerical simulation feasibility study aimed at pointing out preliminary dosimetry issues. Spatial dose distributions incorporating magnetic nanoparticles in MRI-based image-guided radiotherapy have been obtained by Monte Carlo simulation approaches accounting for all relevant radiation interaction properties as well as charged particles coupling with strong external magnetic fields, which are representative of typical MRI-LINAC devices. Two main effects have been evidenced: local dose enhancement (up to 60% at local level) within the infused volume, and non-negligible changes in the dose distribution at the interfaces between different tissues, developing to over 70% for low-density anatomical cavities. Moreover, cellular uptakes up to 10% have been modeled by means of considering different Fe nanoparticle concentrations. A theoretical temperature-dependent model for the thermal enhancement ratio (*TER*) has been used to account for radiosensitization due to hyperthermia. The outcomes demonstrated the reliability of the Monte Carlo approach in accounting for strong magnetic fields and mass distributions from patient-specific anatomy CT scans to assess dose distributions in MRI-based image-guided radiotherapy combined with magnetic nanoparticles, while the hyperthermic radiosensitization provides further and synergic contributions.

## 1. Introduction

Currently, biomedical applications based on nanomaterials are one of the most promising topics at the scientific, industrial, and commercial levels. Nanotechnology advances which enable the production of nanometer-sized objects and nanoparticles (NP) as well as the handling of such have gained significant interest, as their size-dependent physical, chemical, and even biological properties have proved to overcome many biomedical drawbacks.

Within this framework, magnetic Fe nanoparticles (FeNP) are recognized in the biomedicine field [1,2,3,4,5]. Typical cell extension is approximately 10 μm in diameter, whereas cellular internal parts may be within the submicrometric size domain. The controllable size distribution when producing NPs is a relevant advantage for biological applications, as the 10–100 μm range may be used for cells [6,7,8], 0.01–0.5 μm for viruses [9,10], and 0.005–0.05 μm at protein-scale size levels [11,12]. NPs have the potential to interact with biological entities as small probes, suitably accessing complex cellular machinery, and NPs’ coating using biological molecules allows specific interactions/bindings to potentially tag and/or target the biological entity [13,14]. Additionally, FeNPs can be externally manipulated by magnetic fields, thus providing a reliable alternative for FeNP transport and immobilization within patient organs, as well as for tagging biological entities by magnetic means. In addition to this, FeNPs can deliver specific drugs to any targeted region of the body, such as the primary tumor volume (PTV) [15,16,17,18,19], and they can be designed to respond resonantly to time-dependent magnetic fields, providing the potential advantage of energy transfer from the field to the tissues targeted by FeNPs. This physical process is paramount for considering FeNPs as hyperthermia agents capable of producing toxic effects on tumor cells by heating [20]. Magnetic hyperthermia, hereinafter called “hyperthermia”, is included within the category of promising cancer therapy approaches [21]. Traditional techniques based on capacitive heating using a radiofrequency electric field are the most used hyperthermia methods in clinics. However, many features may affect the overall treatment performance of the basic traditional method. Some of the main limiting issues regard tumor target location and extensions, as well as the position and (non) proper adhesion of the electrodes. Therefore, clinical alternatives based on simple heat mediators are preferable to improve the performance of both superficial and deep-located tumors. To this end, magnetic FeNPs were proposed [20,21,22].

According to previous works [23,24], during clinical procedures, FeNPs can be coated by plasma proteins immediately after the injection, allowing the major defense system to recognize the FeNPs. Macrophage cells contribute to FeNP removal, modifying their surface, as a requirement for attaining good biocompatibility, low toxicity, and stability. Therefore, FeNPs need to be stable in water at neutral pH and physiological salinity. Among different potential candidates to be considered as the infusion agent required for the proposed integrated treatment modality, iron-based magnetic nanoparticles, FeNPs, have shown excellent capabilities for theranostics—as well as promising performance in hyperthermia and tumor targeting—along with excellent biocompatibility and surface modification versatility [25]. In vivo FeNP biomedical applications involve both diagnostic and therapeutic purposes as implemented by magnetic targeted delivery of pharmaceutics [26], image enhancement in magnetic resonance imaging (MRI) [27,28], and hyperthermia induced by external magnetic fields [20].

A suitable biological or molecular coating layer is key to interacting with biological targets, acting as an interface between the biological entity and the FeNP. Commonly, biological coatings are antibodies, biopolymers, and monolayers composed by small molecules providing biocompatibility to the FeNPs [29,30,31,32]. Biological molecules, like folic acid, as well as antibodies have shown to well-target superficial cell binding sites and are highly accurate for cell labeling [33]. The coating’s role is not limited to shielding the FeNP; instead, it can be suitably functionalized for attaching different biomolecules [34]. Common FeNP supply is performed by applying intravenous injection directly into the treatment target, or by direct injection of FeNP suspension into the target region. Once administered, FeNPs can diffuse out of bloodstream flow by occupying free intracellular spaces, showing quick results with quite uniform FeNP distributions when using small (~10 nm) FeNPs [35].

Early cancer detection is recognized worldwide as key to a favorable prognosis. MRI and PET/SPECT techniques can identify the disease even before it is anatomically evident. In this context, nanotechnology can play a significant role contributing to one of the major challenges in cancer: accurately establishing the correlations between clinical pathology and the measured cancer biomarkers. Many ongoing studies are devoted to testing different potential FeNP biomedical applications for cancer treatment that include MRI.

Approaches based on ionizing radiation stand among the most effective cancer treatment modalities. Moreover, radiotherapy has advanced by leaps and bounds over the past few decades, introducing modern techniques capable of providing precise conformal dose delivery such as intensity-modulated radiation therapy (IMRT), stereotactic body radiotherapy (SBRT), and, specifically, image-guided radiotherapy, (IGRT) which integrates online imaging during patient irradiation [36,37]. It has been demonstrated the incorporation of cone-beam computed tomography (CBCT) improves radiotherapy delivery precision, thus providing a safe alternative delivery of stereotactic ablative body radiotherapy capable of delivering results comparable with traditional surgery [37,38,39]. However, due to the back-projection reconstruction process, CBCT fails to be fast enough to attain real-time imaging/treatment feedback [37]. On the other hand, MRI can offer improved soft-tissue online imaging for radiotherapy.

In this context, the MRI-LINAC technique, based on the integration of a (1.5 T) MRI scanner with a (6 MV) megavoltage linear accelerator (linac), allows for diagnostic-quality imaging during radiotherapy treatment, enabling real-time adaptive planning/delivery [40,41,42,43]. Nonetheless, there are some drawbacks to be faced when integrating MRI with a linac. From an engineering point of view, MRI uses highly controlled magnetic fields that may interfere with the linac electronics, distorting its shielding. To overcome this problem, the accelerator must be placed outside the Faraday cage, thus isolating both devices [44]. Additionally, and probably more challenging, is that interactions/couplings by the Lorentz force between the MRI magnetic field and charged particles affect dose deposition [45], the “electron return effect” being one of the most remarkable results. In MRI-based techniques aimed at IGRT, such as MRI-LINAC, radiation dose measurements are performed in the presence of strong magnetic fields influencing both the dose distribution as well as the dosimeter response. Therefore, careful investigations are needed to characterize the extent of these influences [46].

During recent decades, several types of contrast agents have been proposed and implemented, aimed at improving the sensitivity of MRI scanning in order to enhance the image information. The performance of magnetic ions and NPs has been remarkably promising [47,48]. The fundamentals for MRI contrast agents regard improving the image quality of tissues and organs by changing the relaxation times of the water protons, thus modifying the signal in the agent-infused body regions [48]. The remarkably promising performance of FeNPs within MRI-based biomedical imaging is demonstrated well by the recent and novel technology called “magnetic particle imaging” that uses oscillating magnetic fields to image superparamagnetic iron oxide NPs as tracers [49] by means of measuring the iron’s change in electronic magnetization, which is more than 100 times stronger than the (standard/reference) signal produced by the change in nuclear magnetization of water protons. Therefore, integrating FeNPs as biomarker/tracer/heat transfer agents with MRI-based IGRT along with hyperthermia radiosensitization may attain significant treatment performance improvements. Similarly, recently developed co-activated catalytic nanoplatforms (CACN) [50] may be considered as potential alternative agents for the proposed integrated treatment, considering the promising performance already shown by the CACN to be MRI imaged while acting for ferroptosis-based synergistic therapies.

Infusing PTV with paramagnetic agents having significant higher atomic numbers (Z) as compared to typical biological tissues should provide selective/preferential accumulation; thus, the paramagnetic properties of FeNPs act as MRI contrast enhancement agents, while the higher atomic number provides higher secondary electron—mainly Auger and Coster–Kronig—production. These short-range electrons (few tens of microns) mainly affect regions close to the NP, thus enabling a local dose reinforcement [51]. High-*Z* NPs have the capacity to enhance targeted ionizing radiation damage in both in vitro and in vivo applications [52,53,54,55,56]. The photoelectric cross-section increases for lower-energy photons as well as for higher-*Z* material types, hence promoting the use of radiological energy range to excite NP agents made of high-*Z* elements [52,56]. The Monte Carlo (MC) method, broadly applied to study radiation transport and interaction, is one of the most reliable and suitable approaches to characterize the effects of NPs interacting with ionizing radiation [56,57]. Different MC codes can be used to estimate energy and dose deposition in different materials, particularly in high-*Z* NP-infused systems [57,58] provided NP size effects are properly accounted for by the MC approach [59]. The use of high-*Z* NPs as radiological biomarkers and radiotherapy enhancers has been extensively studied, assessing how the different de-excitation transitions occur following high-*Z* NP irradiation. Photon (X-ray fluorescence) and secondary electrons, mainly from Auger and Coster–Kronig transitions, are produced; hence, in terms of dosimetry effects, the main consequences of secondary electron production regard short-term (local) dose enhancement [56,59]. Accurate characterization and quantification of such short-term dose enhancement have been very challenging. However, some MC [51,55,57] and experimental [54,57,60] approaches have provided preliminary information about some key issues such as correlation with radiation properties and NP concentration.

Within this framework, a potential combination/integration of an MRI-LINAC-based IGRT technique with hyperthermia and FeNP infusion, a priori, appears to be an attractive opportunity to complement and improve the current performance of cancer theragnostic approaches. This alternative might benefit from the specific advantages of each component, as FeNP infusion might serve, simultaneously, for both multimodal tumor targeting (MRI contrast enhancer and targeting by high-*Z* NP X-ray fluorescence) and local dose enhancement in the tumor, while an extra complementary toxicity contribution to tumor cells may rise from heat transfer during hyperthermia application. Although such an integrated approach is not yet available, feasibility studies and proof-of-concept testing may offer valuable preliminary insights. Thereby, characterizing the dosimetry effects due to the presence of FeNPs in strong magnetic fields, representative of MRI-LINAC like devices, stands out, a priori, as a valuable issue.

The present research focuses on assessing the dosimetry effects in FeNP-infused systems due to the presence of strong magnetic fields, comparable to those used in typical MRI scanners as part of MRI-LINAC-like devices, to estimate the spatial dose distribution distortions in clinical conditions. Accordingly, a MC study has been performed to test the dosimetry effects on FeNP-infused targeted tumors irradiated by a typical LINAC radiotherapy beam, as applied in MRI-LINAC-like devices. Moreover, a CT-based patient-specific approach has been used to attain realistic clinical conditions. A delimited intracranial volume served to model the FeNP-infused target and different iron oxide NP concentrations have been studied. In addition, there are studies performed on different relative directions between the strong magnetic field and the incident radiotherapy beam.

## 2. Results

This section summarizes the most relevant results regarding the proposed methodology to point out the dosimetry effects of FeNPs in MRI-based image-guided radiotherapy. Firstly, the primary photon fluence is reported over-imposed (fused) to the corresponding anatomical section aimed at sketching the proposed methodology by means of assessing differences between fluence, energy, and dose distributions for cases of interest and the reference configuration (no magnetic field, no FeNPs). Then, the charged particle fluence on the corresponding anatomical section is reported to evidence, eventually, the influence of the strong external magnetic field and FeNP infusion. Absorbed dose distribution for the different cases in Table 1 are then reported to highlight effects similar to overall dose enhancement in the PTV target and dose differences at tissue interfaces.

Figure 1 shows an example of the primary particle fluence sagittal distribution for both the reference and the 10% *w*/*w* FeNP concentration and 1.5 T in the X-axis (10-1.5x) cases. The in-depth primary X-ray fluence decreases along the path, following the typical behavior. No significant differences are evidenced due to the external magnetic field, as expected. On the other hand, the presence of FeNPs produces more scattering/absorption processes within the infused (PTV) region, thus slightly decreasing the primary photon fluence beyond the PTV.

Figure 2 and Figure 3 report the secondary charged particle (electron) fluence distribution for the 1% *w*/*w* FeNP and 0 T (1-0), 1% *w*/*w* FeNP and 1.5 T in the X-axis (1-1.5x), and 10% *w*/*w* FeNP and 1.5 T (10-1.5x) cases. As can be appreciated, significant effects on the charged particle fluence derive from the presence of the strong external magnetic field, and these effects are clearly enhanced close to the intracranial cavity’s interfaces, mainly due to the so called “electron return effect” that refers to the high influence of the Lorentz force changing the electron track curvature in low-density media. The secondary charged particle fluence concentration—colored in red, within the oral–nasal cavity at z = 105 pix, approximately, clearly evidenced in Figure 2b and Figure 3b—is a typical example of the magnetic field influence. Moreover, charged particle fluence strongly determines the further absorbed dose; thus, similar effects might be expected on absorbed energy/dose levels.

Figure 4 shows a representative dose distribution example, reporting the transversal dose map for the 1% *w*/*w* and 1.5 T in the *x*-axis direction (1-1.5x) and reference cases along with the corresponding dose profiles within a region of interest (ROI) around the MNP-infused target.

According to results reported in Figure 4, the volume corresponding to the FeNP-infused PTV target is completely covered by the incident X-ray beam, regardless of the presence and direction of the magnetic field and the FeNP concentration. Nonetheless, the magnetic field modifies the absorbed dose distribution, which can be appreciated in the vertical/horizontal dose profiles. This effect is mainly attributable to the Lorentz force acting on the secondary charged particles. As is clearly shown, the transversal *y* (horizontal) dose profile depicts how the absorbed dose is affected according to the charged particles’ direction. The Lorentz force acts, producing opposite torque for charged particles flowing right or left, positive/negative polar angle, from the +*z*-axis. On the other hand, because the magnetic field is applied in this direction, the transversal *x* (vertical) dose distribution is hardly affected.

Aimed at providing a complete overview on the dosimetry effects close to the FeNP-infused PTV target, Figure 5 reports the absorbed energy distribution profiles for an ROI around the target for the transversal plane, varying FeNP concentration with and without perpendicular (*x*−) magnetic field. As may be appreciated, non-negligible dosimetry effects due to the presence of the perpendicular (*x*−) magnetic field are obtained for the horizontal (*y*) profile as a direct consequence of the Lorentz force. Additionally, the FeNP concentration has no significant influence in dose distortions as compared with the reference case, pointing out that absorbed energy distribution changes are mainly a result of the presence of a magnetic field. 

Similarly, Figure 6 reports the dose distribution profiles for an ROI around the target for the transversal plane, varying FeNP concentration corresponding to with and without parallel (*z*−) magnetic field. As observed, in the case of a parallel (*z*−) magnetic field, the influence of the FeNP concentration is significant for the transversal *y* (horizontal) dose profile, whereas the magnetic field does not strongly affect the transversal dose profiles. 

It is worth mentioning that results shown in Figure 5 and Figure 6 have been confirmed by absorbed energy distributions, thus avoiding potential distortions from analytic estimation of the compound (magnetic FeNP infusing soft tissue) mass density.

Figure 7 reports the *DER* depicting the mean magnetic FeNP-infused PTV (target) absorbed dose, and corresponding uncertainty, for different FeNP concentrations, with and without magnetic field.

As observed, noticeable dose increases are attainable by infusing the PTV target volume with FeNPs, and this dose reinforcement is correlated with the FeNP concentration regardless of the presence of the magnetic field and its relative orientation with respect to the incident X-ray beam. Quantitative results reported in Figure 7 depend on the compound density model used to assess the mean mass density of the magnetic FeNP-infused tumoral tissue. Nevertheless, the overall trend confirming the dose reinforcement effect represents the more relevant issue to be remarked upon.

Finally, the integral dose enhancement (*IDE*), integrating the dose enhancement for the proposed nano-thermo-radiotherapy (*IDE_nTRT_*) due to the presence of magnetic FeNPs with hyperthermia radiosensitization, as proposed by expression (2), can be calculated for the different FeNP concentrations by incorporating the corresponding *TER* for the glioblastoma PTV once the hyperthermia temperature *T* and application time *t* are established. Figure 8 reports the obtained results for the *IDE* corresponding to the FeNP-infused glioblastoma PTV considering three scenarios: (i) T = 39 °C for 60 min, (ii) 42 °C for 90 min, and (iii) no hyperthermia (Figure 7 scenario), for both orientations of the magnetic field with respect to the optical axis.

## 3. Discussion

The spatial distribution of primary X-ray fluence, reported in Figure 1, depicted the proposed methodology, reporting the physical/radiological quantities fused to the corresponding patient anatomy. Furthermore, the radiological consequences from the irradiation configuration, shown in Figure 9 in the Materials and Methods section, can be appreciated. As expected, no influences on the external magnetic field arose from the primary X-ray (non-charged particle, photon) fluence. On the other hand, the presence of FeNPs infusing the PTV target have shown to affect the primary X-ray fluence due to the different (increased) cross-section for higher FeNP concentrations.

In addition, the charged particle fluence is significantly affected by the presence of the external strong magnetic field, as shown in Figure 2 and Figure 3. Although both electrons and positrons are created as secondary charged particles when irradiating with a 6 MV beam, it is worth noting that electron fluence is significantly higher than that of positrons. In fact, electron fluence represents an acceptable approximation of the (total) charged particle fluence. Additionally, the charged particle fluence is strongly dependent upon the magnetic field orientation, while major differences in contrast to the reference (no magnetic field) case are observed at material interfaces, mainly those involving low-density media (lung, air cavities, etc.). Regarding the effects of the presence of FeNP, the higher the concentration, the larger secondary particle production, thus providing a local dose increase.

Finally, effects on the absorbed dose distribution, both due to the magnetic field and the FeNP infusion, arose mainly from changes in the charged particle fluence. As expected, the relative orientation of the magnetic field and the incident X-ray beam is a key issue. Furthermore, any arbitrary orientation can be suitably described in terms of the corresponding components, perpendicular and parallel to the incident X-ray beam. The overall dose difference around the region of interest, as depicted in Figure 8, suggests that an extra dose contribution is obtained according to the FeNP concentration.

Within this context, potential theragnostic approaches integrating MRI-based IGRT with magnetic hyperthermia may benefit from accurate dose delivery because of the real-time and precise IGRT provided by the MRI along with extra dose contributions coming from secondary charged particles produced by infusing the PTV with FeNP, as well as the biological damage to FeNP-targeted tumoral cells provided by the hyperthermia. Although modeling of hyperthermic effects is beyond the purposes of the present study, it is worth stating that a successful integrated hyperthermia application may strongly depend on the ability to target the tumor cells by the selective uptake of NP-based linkers. Similarly, transporting the magnetic NP-based agent to the required target volume to be treated—PTV, for instance—represents another issue to be addressed. Thus, aimed at potential clinical applications, high-*Z* NP concentration should be kept within low-toxicity levels, require non-expensive synthesis, and present a high-saturation magnetic moment; therefore, magnetite (Fe_3_O_4_) might constitute a candidate.

As it is known, tumors tend to have hypoxic cells within the inner part of the tumor bulk and these cells are high-ionizing radiation resistant; however, they may be very sensitive to heat. Consequently, integrating hyperthermia with modern accurate radiotherapy might provide for high-oxygen cell killing by oxygenating inner hypoxic tumoral cells with ionizing radiation complemented with hyperthermia, thus improving the ionizing radiation damage and the overall treatment performance. Although the specific modality to synergistically integrate hyperthermia with radiotherapy needs to be studied, preliminary results for superficial tumors have shown to yield higher complete and durable responses than ionizing radiation alone, whereas research is still in progress for deep tumors [61,62]. Hyperthermia has proved to mainly improve overall radiotherapy treatment by reducing neoplastic cell survival, and also by reducing the normal tissue complication probability (NTCP) due to inhibition of the expression of radiation-induced damage in normal cells [62,63]. Moreover, studies conducted in animals have demonstrated significant tumor regression in mouse epidermoid carcinoma xenograft by preliminarily combining radiotherapy with hyperthermia using coated NPs [64,65]. According to the results reported in Section 3, the presence of a strong external magnetic field—as required for MRI-based IGRT—along with Fe-based nanoparticle (FeNP) infusion to the PTV produced distortions in the reference (no magnetic field, no FeNP infusion) absorbed dose pattern, which are evidenced at centimeter-scale and are particularly significant at the interfaces involving low-density materials, such as internal air/lung cavities for the intracranial region. These findings agree with previous studies [44,65]. Moreover, the deposited energy/dose enhancement has been found to be, a priori, linearly correlated with the FeNPs when irradiating with a 6 MV photon beam, thus, coherent with results reported in Figure 8 for the low FeNP concentration range. The remarkable dosimetry effects due to the presence of a strong external magnetic field at the material interfaces have also been reported by other authors [44,66].

Nano-thermo-radiotherapy by synergic combination of magnetic nanoparticles in MRI-guided radiotherapy and hyperthermia radiosensitization potentially represents a significant and disruptive alternative as well as an improvement to the current biomedical applications. To this aim, a suitable dosimetry model appears as a key issue and the approach herein described is shown to be auto-consistent and suitable at providing a first-order approximation of the overall potential dose reinforcement, as reported in Figure 8.

Once demonstrated that the presence of FeNPs infusing a PTV region using non-toxic concentrations provided an extra contribution to the total absorbed dose because of the secondary short-range charged particle production, it seems suitable to evaluate alternative novel approaches to integrate hyperthermia with MRI-based IGRT. Some of the potential benefits regard: (i) determination of FeNP distribution/uptake by means of MRI, (ii) guiding the FeNPs using magnetic fields, (iii) increasing tumoral cell radiosensitivity by hyperthermia, mainly for the hypoxic inner bulk cells, and (iv) attaining two extra dose contributions due to the presence of FeNPs—a local dose enhancement coming from the secondary charged particles produced by ionizing radiation, and another biological tumor damage provided by the hyperthermic heating.

## 4. Materials and Methods

In this study, theoretical and Monte Carlo approaches have been employed to assess the dosimetry effects due to the synergy from FeNPs and hyperthermia as radiosensitizer. To this aim, expression (1) summarizes the main concept of the proposed approach:
*IDE* = ***f***(*D_Ref_, D_HT_, D_FeNP_*)
(1)

where *IDE* is the integral dose enhancement representing the overall dose reinforcement, which is characterized as a function of the contributions from hyperthermia (*D_HT_*) and the presence of FeNPs (*D_FeNP_*) as well as the reference dose (*D_Ref_*) that corresponds to the absorbed dose value without hyperthermic radiosensitization and in the absence of FeNPs. Although the exact form of the ***f*** function might be complex to assess, some general characteristics and trends can be approached. For instance, expression (2) is obtained assuming first-order independent variable separation between hyperthermic and FeNP effects; thus, the convolution, denoted by the ⊗ symbol, between both contributions (***f_HT_*** and ***f_FeNP_***) can be approached as the direct arithmetic product (●) of independent coefficients [67]:
*IDE* = ***f***(*D_Ref_*, *D_HT_*, *D_FeNP_*) = ***f_HT_***(*D_Ref_*, *D_HT_*) ⊗ ***f_FeNP_***(*D_Ref_*, *D_FeNP_*) ~ ***f_HT_* ● *f_FeNP_***
(2)


Finally, the effects due to FeNPs (***f_FeNP_***) can be modeled—at least in a first-order approach—as the dose enhancement radio (*DER*) [66], whereas the thermal enhancement ratio (*TER*) [67,68,69,70] can be used to account for the hyperthermic radiosensitization contribution (***f_HT_***).

### 4.1. Dosimetry Effects Due to FeNPs Characterized by Monte Carlo Simulations

Dedicated subroutines were adapted from a well-known and validated main code (FLUKA) to model the radiation transport and collision in simulation configurations incorporating strong magnetic fields and the presence of FeNPs. Patient-specific anatomy was incorporated by means of a voxelization approach enabling the simulation to assign mass distribution according to the patient CT. Aiming at evaluating the preliminary feasibility of obtaining local dose enhancement by infusing FeNPs, spherical-shaped FeNPs—distributed in size according to a normal (Gaussian) distribution—were modeled considering a 10 nm diameter mean value with FWHM of 3 nm. Different FeNP concentrations were used to fill a small (~1 cm^3^) target volume, a realistic 6 MV X-ray beam was defined, and, finally, the absorbed dose distribution for different configurations regarding the relative magnetic direction between the magnetic field and the incident radiotherapy beam was calculated.

The radiation transport, accounting for scattering and absorption interactions, is described by the main Boltzmann transport equation (BTE) [60]. However, analytic solutions to the BTE are commonly difficult to achieve or non-available in most practical cases. On the other hand, numerical methods have proven to be reliable and suitable BTE-solving alternatives in complex situations, such as considering non-homogeneous media or regions consisting of complex boundaries. In fact, MC techniques based on stochastic methods are probably the most promising and suitable approach to performing full radiation transport in complex systems [71,72].

Therefore, effective MC methods are nowadays implemented by many different main codes, such as GEANT [71], PENELOPE [72], EGS [73], MCNP [74], and FLUKA [75,76], that have demonstrated a capability for simulating accurate radiation transport for the study of dose distributions in complex irradiation configurations, as required for medical physics applications such as providing accurate descriptions in radiotherapy [53,58,77,78,79], radiology [56,57,80], and nuclear medicine [81,82,83], among other medical physics purposes.

#### 4.1.1. Monte Carlo Model Accounting for External Magnetic Field

The FLUKA main code has been largely and successfully tested to assess radiation transport in complex systems, even accounting for electromagnetic fields [84,85,86] and handling NP-infused systems [57,87,88]. The FLUKA code allows the user to define or create virtual materials by assigning the required physico-chemical properties, such as mass density, element constitution, and mean excitation energy. Similarly, the simulation geometry, primary particle properties, and overall simulation setup can be defined by means of the FLAIR graphical user interface. This tool is useful to configure simulation sketches involving NP-infused systems, providing, for example, reliable descriptions of local dose enhancement.

The presence of external magnetic fields should not affect the energy transferred to secondary particles (Kerma) following interactions for steady state (constant primary fluence) conditions. However, the Lorentz force influences the charged particle (electrons and positrons in typical radiotherapy conditions) kinematics, as illustrated in Figure 9.

It should be noted that under these conditions, electro-magnetic field effects can be treated independently from that of the radiation–matter interactions. Moreover, in the typical radiotherapy energy range, the proportion of secondary positrons is significantly smaller than that of secondary electrons.

The basis for the charged particle tracking in the presence of strong magnetic fields may be briefly summarized as follows: charged particles may change (loose) kinetic energy and traveling direction due to interactions. In standard MC simulations (absence of electro-magnetic fields), charged particles travel the distance freely, as a straight trajectory segment, up to the next interaction location according to the corresponding stopping power (*S*). On the contrary, when external electro-magnetic fields are considered, the approach consists of assuming that radiation-interaction properties are not significantly affected, thereby accounting for the electro-magnetic field by modifying the charged particles’ tracking according to the Lorentz force ***F*** as depicted for the free space case by Equation (3):***F*** = z_e_q_e_ [***E*** + 1/c ***v*** × ***B***](3)
where *q_e_* is the electron element charge, *z_e_* equals +1 for positrons (e^+^) and equals −1 for electrons (e^−^), *c* is the speed of light in vacuum, ***v*** the charged particle speed, and ***E*** and ***B*** are the electric and magnetic field, respectively. Thus, in the absence of electric fields, the Lorentz force strength depends only on the particle velocity component perpendicular to the magnetic field. Therefore, when charged particles travel through vacuum at an angle α between the external magnetic field and the velocity, they will follow a helical trajectory. Since the secondary electrons produced by typical megavoltage X-ray radiotherapy beams predominantly tend to have relatively small deviations with respect to the incident beam, α assumes low values when the external magnetic field and the incident X-ray beam are parallel, whereas the helical trajectory radius increases for the orthogonal configuration (α~π/2). It is worth mentioning that the Lorentz force does not change the charged particles’ kinetic energy if synchrotron radiation losses are neglected, which are in fact orders of magnitude lower than the kinetic energies involved [85,89]. In summary, the interactions and collisions of the ionizing radiation with biological tissues and FeNPs are modeled by standard Monte Carlo radiation–matter interaction approaches, whereas particle transport/tracking is modified by including the Lorentz Force, as described by expression (3).

#### 4.1.2. Simulation of Realistic Patient-Specific Setup

Attaining reliable dosimetry in realistic clinical conditions requires patient-specific approaches during treatment planning dose calculation. In this regard, three-dimensional (3D) voxel-level dosimetry stands out as a major improvement. The FLUKA code can import patient-specific information at the voxel level by means of CT data [87,90]. An external ascii user-defined file serves to establish the conversion between the CT Hounsfield Unit (HU) and the corresponding simulation material. Moreover, dedicated adaptations of the *magfld* and *source* FLUKA routines allow one to suitably define user-desired external magnetic fields and primary particle properties, respectively [86,91].

An anonymized head–neck CT was used to achieve typical clinical conditions along with primary photons distributing energy and direction according to typical (ELEKTA) 6 MV linac phase spaces [66], as reported by Figure 10 for the spectral distribution.

Cut-off values for radiation transport were fixed at 0.001 MeV in all cases. Simulations have been performed on cluster facilities computing five independent cycles of 10^8^ primary particles for each one to ensure statistical uncertainties of less than 2% in all cases. Figure 11 depicts the simulation setup as shown by *FLAIR*, the FLUKA visualization interface.

To show the effects of strong magnetic field presence and/or FeNPs infusing the target, the absence of both magnetic field and FeNP was considered as the reference case, whereas all other configurations were treated as testing cases potentially affected by the field and/or the FeNPs. The PTV target was placed directly on the incident X-ray path, approximately 3 cm depth, and delimited (in pixels) by: [250 ≤ x ≤ 260] × [140 ≤ y ≤ 150] × [150 ≤ z ≤ 160]. Different FeNP concentrations were studied by defining virtual compounds modeling soft tissue containing different Fe mass concentrations ranging from [0, 10]%. In addition, a 1.5 T strength was considered for the external magnetic field, as is representative of typical MRI-LINAC-like technologies. Table 1 summarizes the different configurations studied.

Relative variations of the dose distributions have been calculated as expressed by Equation (4): Δ*D*(***r***) = 100%∙(D(***r***) − D_Ref_ (***r***))/D_Ref_ (***r***)(4)
where Δ*D*(***r***) represents the dose difference at the ***r*** position as obtained from any (test) dose map (*D*(***r***)) and the corresponding reference dose distribution (*D_Ref_* (***r***)). This way, the dose enhancement ratio *DER* (***f_FeNP_*** in expression (2)) is obtained as the quotient between the absorbed dose in the presence of FeNP and the absorbed dose in reference conditions, i.e., without FeNPs. For completeness, overall uncertainties have been assessed by MC statistical uncertainty outputs and error propagation applied to Equation (2). Finally, the absorbed dose mean value and uncertainty for the FeNP-infused volume has been calculated restricting the ***r*** values to those voxels within the targeted volume. Home-made MatLab^®^-supported scripts have been developed and implemented to process and visualize MC outputs fused/super-imposed to the corresponding anatomic CT slice.

### 4.2. Thermal Radiosentization Model

The model implemented for TER (**f_HT_**) is based on representing the biological iso-effects as characterized by means of the linear–quadratic (LQ) radiobiological model for cell survival/viability (*S*) [92]. Thus, explicit dependencies on the temperature *T* and time *t* of the hyperthermia application are required. To this aim, the adopted model consists of considering explicit dependence of the LQ parameters (*α*, *β*) on *T* and *t* as shown in (5):−*ln*(*S*) *= α*(*T*, *t*) *D* +*β*(*T*, *t*) D^2^(5)

The functional dependence of *α* and *β* on the hyperthermic temperature *T* and time *t* can be introduced according to theoretical and/or empirical approaches.

Both theoretical and experimental approaches indicate that *TER* = *D_HT_/D_Ref_* varies—at least as first-order—linearly with the elapsed hyperthermia application time, whereas basic thermodynamics [93] can be used to show that the temperature-dependence can be obtained by considering the reaction energy absorbed in the transition at the constant rate of the heat capacity, i.e., the Gibbs free energy [94]. In this context, the TER can be assessed by expression (6):*TER* = *TER*_0_(*T_Ref_*, *t* = 0) + c_1_ t e^c2(T−Tc)^(6)

The coefficients *c*_1_ and *c*_2_ need to be fitted from experimental data according to each cell type, while *T_c_* represents the melting point [70]. Then, the first term in (6) is the offset value corresponding to the reference case. Figure 12 reports a typical example depicting the *TER* multivariable dependence.

Moreover, experimental results have largely confirmed this trend for many different cell types [64]. Although the glioblastoma multiforme (GBM), a grade-4 astrocytoma, is among the most aggressive brain tumors, hyperthermic effects have been confirmed for GBM as an effective means of radiosensitization for further treatment [95,96]. According to the standard LQ model, radiosensitization effects are correlated with an *α*/*β* ratio, which attains larger values for more radioresistant cells. Conversely, in the proposed framework, the *α*(*T*, *t*)/*β*(*T*, *t*) ratio is reduced due to sub-lethal thermal damage to account for the hyperthermia effects. Figure 13 summarizes the viability of glioblastoma cells after hyperthermia at different application temperatures along with comparisons with the theoretical model by fitting expression (6) to experimental data.

As may be appreciated, remarkable agreements are found between theory and experiments, as can be noticed by the *R*^2^ correlation parameter—reported in Table 2—remaining close to 1 in all cases.

## 5. Conclusions

Dosimetry effects due to magnetic FeNP infusion in the presence of strong magnetic fields were assessed by means of the proposed Monte Carlo approach. Moreover, realistic clinical conditions were attained by introducing a patient-specific anatomy, modeling the 6 MV photon beam by means of its corresponding phase space properties, introducing a 1.5 Tesla magnetic field, and defining FeNP-infused PTV regions considering non-toxic concentrations.

The overall dosimetry effects due to the synergic combination of magnetic FeNP under MRI-guided radiotherapy and hyperthermia—as described by the nano-thermo-radiotherapy integral dose enhancement (*IDE_nTRT_*)—have been satisfactorily characterized, at least by a first approximation attempt.

The proposed methodology was applied to intracranial radiotherapy using patient CT anatomy providing the spatial dose distribution differences between 12 configurations of interest and the reference case (no magnetic field, no FeNPs), highlighting remarkable effects at material interfaces involving low-density media, such as the oral and nasal cavities for the head and neck region. Moreover, obtaining PTV dose increase according to the infused FeNP non-toxic concentration constituted a relevant finding. The implementation of the proposed model to a patient-specific situation has shown, preliminarily, promising performance in modeling the synergic dosimetry effects from high-*Z* NP and hyperthermia radiosensitization considering realistic approaches for both the anatomy and the MRI magnetic strengths as applied to a glioblastoma multiforme. The proposed model to characterize the integral nano-thermo-radiotherapy integral dose enhancement (*IDE_nTRT_*) may contribute to the ongoing progress in the field of combining FeNP infusion in MRI-guided radiotherapy with hyperthermia radiosensitization as a potential dosimetry tool to contribute to research and clinical assays to be trialed.

This research focused on the dosimetry effects due to non-toxic magnetic nanoparticle infusion in the presence of strong external magnetic fields as a preliminary step to address the reliability of integrating hyperthermia with the modern technology of MRI-based image-guided radiotherapy.

## Figures and Tables

**Figure 1 ijms-24-00514-f001:**
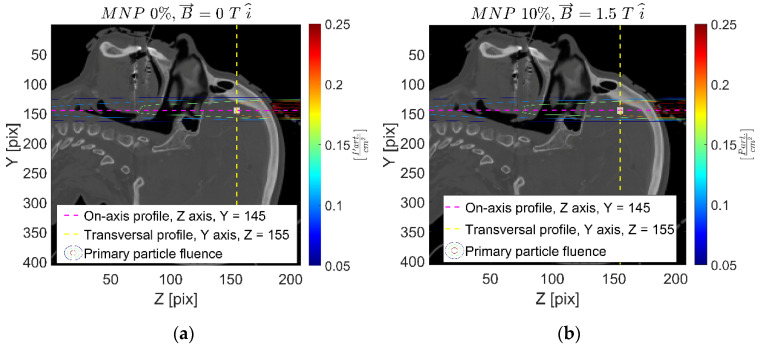
Primary particle fluence map on sagittal plane showing magnetic FeNP-infused target volume and vertical/horizontal profiles (light-/dark-colored dash lines) along with level contours at 25% (blue), 50% (green), and 75% (orange) of the maximum absorbed dose for the reference (**a**) and 10% *w*/*w* FeNP with 1.5 T (10-1.5x) (**b**) configurations. Optical (central) axis of the primary X-ray beam corresponds to the dark horizontal dash line.

**Figure 2 ijms-24-00514-f002:**
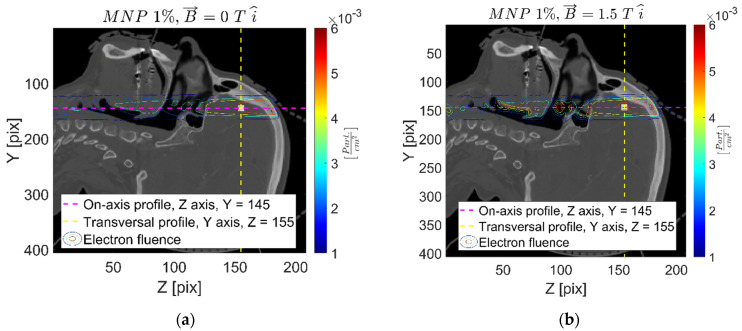
Secondary charged particle fluence map, sagittal plane, for the 1-0 (**a**) and 1-1.5x (**b**) cases.

**Figure 3 ijms-24-00514-f003:**
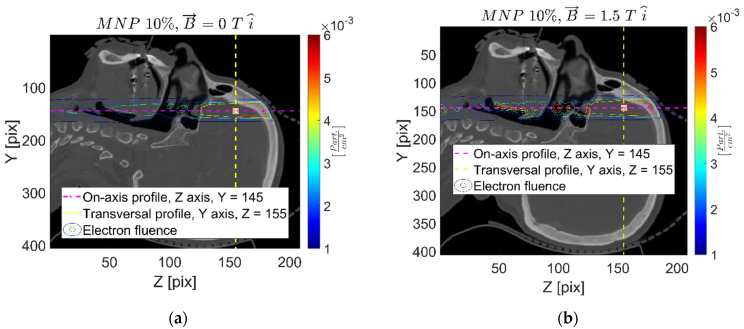
Secondary charged particle fluence map, sagittal plane, for 10-0 (**a**) and 10-1.5x (**b**) cases.

**Figure 4 ijms-24-00514-f004:**
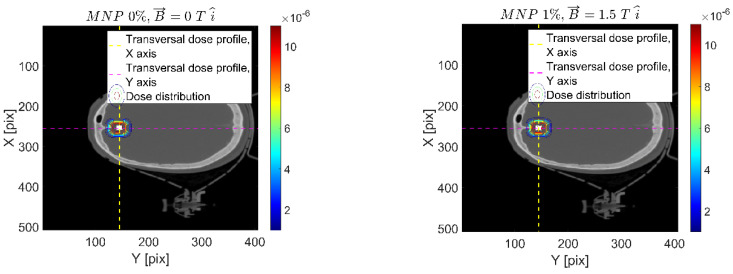
Transversal dose distribution maps (top) for the reference (**a**) and 1% *w*/*w* FeNP with 1.5 T in *+x* (1-1.5x) (**b**) configurations along with corresponding profile comparisons (bottom). Magnetic FeNP-infused targeted volume (light-colored). Absorbed dose is reported on arbitrary unit [GeV/g per primary particle] scale.

**Figure 5 ijms-24-00514-f005:**
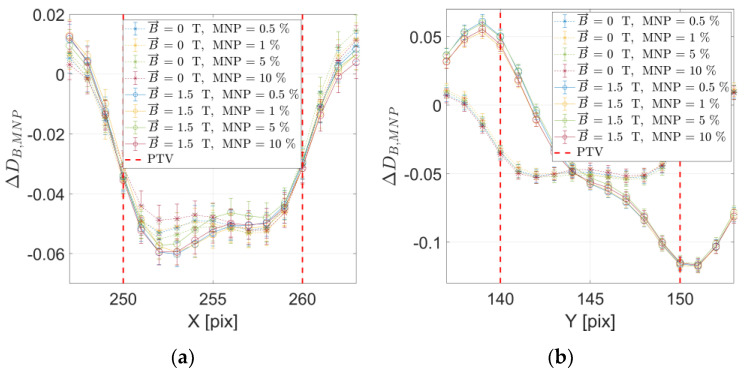
Dose distribution vertical (**a**) and horizontal (**b**) profiles for the transversal plane reporting all configurations, varying FeNP concentration and magnetic field (perpendicular to incident X-ray beam) presence. Vertical red dashed lines delimit the MNP-infused target (PTV). Reported uncertainties correspond to 1 standard deviation.

**Figure 6 ijms-24-00514-f006:**
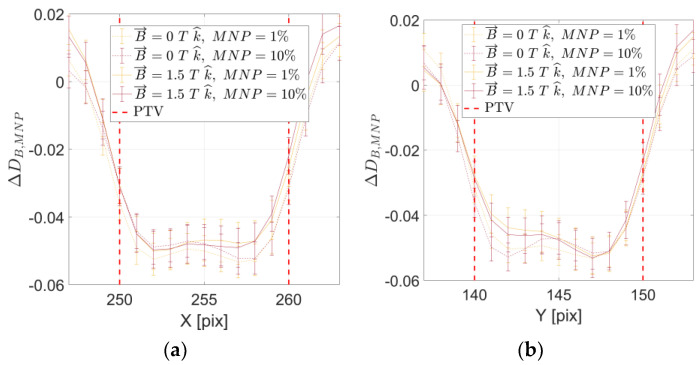
Dose distribution vertical (**a**) and horizontal (**b**) profiles for the transversal plane reporting all configurations, varying MNP concentration and magnetic field (parallel to incident X-ray beam) presence. Vertical red dashed lines delimit the FeNP-infused target (PTV). Reported uncertainties correspond to 1 standard deviation.

**Figure 7 ijms-24-00514-f007:**
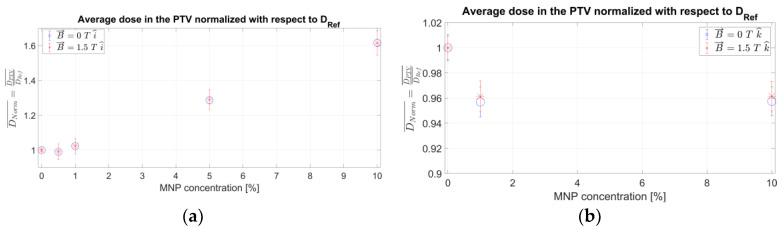
Mean absorbed dose within the PTV target volume, normalized to the reference (no magnetic field, no magnetic FeNP infusion) case, for different FeNP concentrations in absence of external magnetic field and incorporating 1.5 T magnetic field perpendicular (**a**) and parallel (**b**) to the incident X-ray beam.

**Figure 8 ijms-24-00514-f008:**
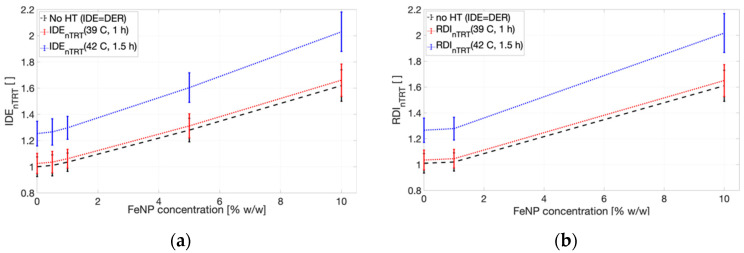
Nano-thermo-radiotherapy integral dose enhancement (*IDE_nTRT_*) at the glioblastoma PTV as a function of the different FeNP concentrations for a 1.5 T magnetic field perpendicular (**a**) and parallel (**b**) to the incident X-ray beam.

**Figure 9 ijms-24-00514-f009:**
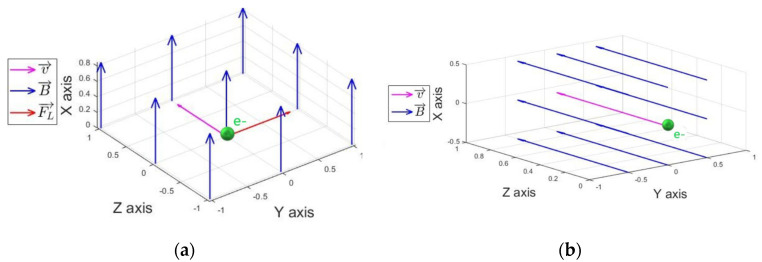
Illustration of charged particle tracking with Lorentz force due to magnetic field perpendicular (**a**) and parallel (**b**) to the radiation beam.

**Figure 10 ijms-24-00514-f010:**
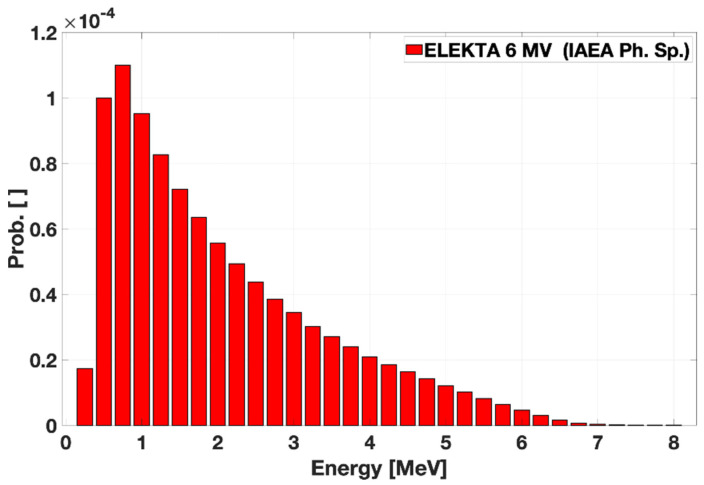
ELEKTA 6 MV photon spectrum as extracted from the IAEA phase space.

**Figure 11 ijms-24-00514-f011:**
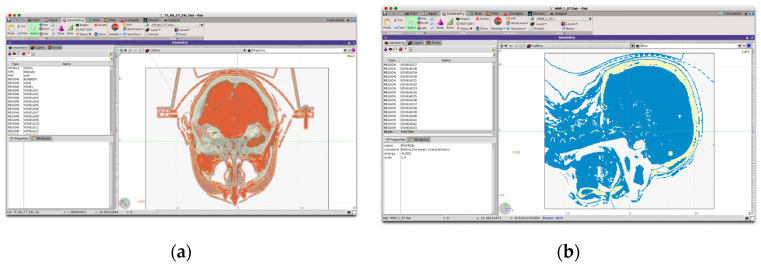
FLAIR simulation interface showing head–neck CT geometry on different planes (**a**) and the irradiation setup depicting the incident X-ray beam along the −*z* direction and the FeNP-infused (light-colored in frontal skull cortex) target volume (**b**).

**Figure 12 ijms-24-00514-f012:**
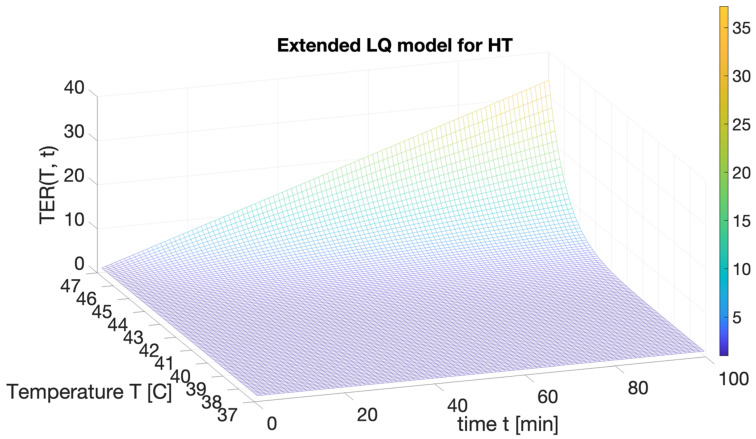
Typical illustrative example of the thermal enhancement ratio *TER* as a function of the hyperthermia temperature *T* and time *t* using 37 degrees Celsius as the reference condition.

**Figure 13 ijms-24-00514-f013:**
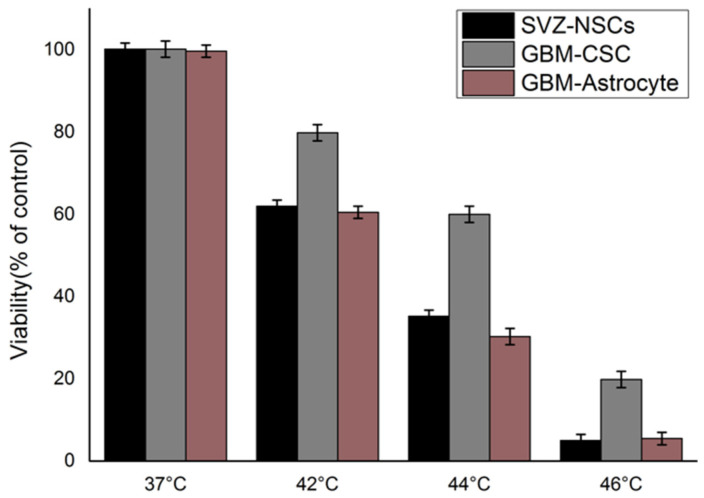
Glioblastoma cell survival rate for the SVZ, CSC, and astrocyte types applying different temperatures for 60 min (data from [96]) (top) and corresponding regression fits by means of (6) (bottom).

**Table 1 ijms-24-00514-t001:** List of studied configurations for the X-ray beam flowing along the negative (decreasing) *z*-axis direction.

Setup	FeNP Mass Concentration [%]	Magnetic Field [T]
Setup 0-0 (Reference)	0	(0,0,0)
Setup 0-1.5x	0	(1.5,0,0)
Setup 0-1.5z	0	(0,0,1.5)
Setup 0.5-0	0.5	(0,0,0)
Setup 0.5-1.5x	0.5	(1.5,0,0)
Setup 1-0	1	(0,0,0)
Setup 1-1.5x	1	(1.5,0,0)
Setup 1-1.5z	1	(0,0,1.5)
Setup 5-0	5	(0,0,0)
Setup 5-1.5x	5	(1.5,0,0)
Setup 10-0	10	(0,0,0)
Setup 10-1.5x	10	(1.5,0,0)
Setup 10-1.5z	10	(0,0,1.5)

**Table 2 ijms-24-00514-t002:** Fitting correlation factor *R^2^* for the different glioblastoma cells.

SVZ-NSC	GBM-CSC	GBM-Astrocyte
0.99944	0.99978	0.99698

## Data Availability

Not applicable.

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
