# Peer review of "Dosimetry Effects Due to the Presence of Fe Nanoparticles for Potential Combination of Hyperthermic Cancer Treatment with MRI-Based Image-Guided Radiotherapy"

_ijms, 2022, doi:10.3390/ijms24010514_

Round 1

Reviewer 1 Report

This work describes the dosimetry effects due to non-toxic magnetic nanoparticle FeNPs infusion in the presence of strong external magnetic fields, as a preliminary step to address the reliability of integrating hyperthermia with the modern technology for MRI-based image guided radiotherapy.It is quite complete and uses original and sophisticated methodologies. I propose publication of the manuscript as it is.

Author Response

Dear Reviewer, Please notice that authors have provide a point-by-point answer list to all the Reviewers' queries. Considering the suitability of the required corrections as well as the provided suggestions, authors have prepared a unique document addressing, both integrally and one-by-one the Reviewers' queries. So, please find in the attached document the answers to your queries and comments. Once again, thank for your valuable review and insights.

Reviewer 2 Report

In this manuscript, the authors demonstrated the method for evaluating MRI-based image guided radiotherapy in combination with the dose distribution of magnetic nanoparticles. This paper involves a lot of simulation and formula calculation, such as Monte Carlo simulation. Many experiments have done to prove the statement. However, the following major points would be required prior to publishing in a journal of the International Journal of Molecular Sciences:

1.      In the article, all the experiments are based on Fe nanoparticles. However, there are many kinds of Fe nanoparticles, what’s the size of the Fe nanoparticles you used? How about the morphology and performance in vivo? Could you provide some data by using Dynamic light scattering or Transmission Electron Microscope?

2.      In the introduction, the authors mentioned that “During clinical procedures, FeNPs are coated by plasma proteins immediately after the injection, allowing the major defense system to recognize the FeNPs.”. What’s the surface modification of FeNPs to improve biocompatibility in this manuscript? If the FeNPs you used can be coated by plasma proteins in vivo immediately, please provide more experiments to prove the biocompatibility and cite authoritative literature for proof.

3.      Other Fe theranostic nanoplatform may exhibit similar functions for MRI guided cancer therapy,   (Chem, 2022, Volume 8, Issue 7, 1956-1981, Angew. Chem. Int. Ed. 2021, 60, 9562-9572, Nature Biomedical Engineering 2020,4, 325, Nano Lett. 2018, 18, 182. Angew. Chem. Int. Ed. 2022., https://doi.org/10.1002/ange.202206074), what are the unique advantages for Fe nanoparticles? The author should discuss the intrinsic advantages for Fe nanoparticles, in contrast to other works. 

4.      Why the FeNPs can be used for MRI-based image guided radiotherapy? What’s the meaning of the high Z elements? Can FeNPs be defined as high Z elements?

5.      In this paper, FeNPs also can be used for magnetic hyperthermia. However, the association between radiotherapy and hyperthermia may require more data.

6.      Some information may be missing or obscured in figure 12.

7.      Some figures are not easy to distinguish. Can you enlarge the different areas in partial pictures?

Author Response

(The authors gave the same response as above.)

Reviewer 3 Report

Dear authors and editors, I have carefully read the manuscript. The authors use an interesting idea, but with few clinical applications. The study is only theoretical and with MC simulations.

The manuscript is of medium scientific value. The impact is mediocre and contains no errors. I believe that IJMS is a broadly applicative journal. The topic of this manuscript is totally off-topic with the Special Issue and with the "Section - Molecular Oncology" I think journals such as Applied Sciences or others are more appropriate.

The main focus (Nanoparticles? Medical Imagin? Radio-therapy?) is unclear.

Therefore I do not recommend the manuscript for publication. I don't criticize the quality, but the editorial position.

Author Response

(The authors gave the same response as above.)

Round 2

Reviewer 3 Report

I have read the manuscript carefully.

The authors made no significant changes.

My doubt about the editorial placement remains.

I suggest a move with subsequent acceptance without further refereeing at other MPDI Journals. Applied Sciences or JPM.